# SketchEdit: Editing Freehand Sketches at the Stroke-level

## Abstract

Freehand sketching is a representation of human cognition of the real world. Recent sketch synthesis methods have demonstrated the capability of generating lifelike outcomes. However, these methods directly encode the whole sketch instances and makes it challenging to decouple the strokes from the sketches and have difficulty in controlling local sketch synthesis, e.g., stroke editing. Besides, the sketch editing task encounters the issue of accurately positioning the edited strokes, because users may not be able to draw on the exact position and the same stroke may appear on various locations in different sketches. We propose SketchEdit to realize flexible editing of sketches at the stroke-level for the first time. To tackle the challenge of decoupling strokes, our SketchEdit divides a drawing sequence of a sketch into a series of strokes based on the pen state, align the stroke segments to have the same starting position, and learns the embeddings of every stroke by a proposed stroke encoder. This design allows users to conveniently select the strokes for editing at any locations. Moreover, we overcome the problem of stroke placement via a diffusion process, which progressively generate the locations for the strokes to be synthesized, using the stroke features as the guiding condition. Both the stroke embeddings and the generated locations are fed into a sequence decoder to synthesize the manipulated sketch. The stroke encoder and the sequence decoder are jointly pre-trained under the autoencoder paradigm, with an extra image decoder to learn the local structure of sketches. Experiments demonstrate that the SketchEdit is effective for stroke-level sketch editing and outperforms state-of-the-art methods in the sketch reconstruction task.

## 1 Introduction

People may draw sketches to express their abstract concepts for the real world, and humans possess an extraordinary ability to create imaginative sketches. The objective of sketch synthesis is to mimic the human drawing process through machines, and the task is challenging due to the sketch abstractness, sparsity, and lack of details. Recently, efforts have been made to learn efficient sketch representations and generate realistic sketches, such as Sketch-RNN (Ha & Eck, 2017), SketchLattice (Qi et al., 2021), SketchHealer (Su et al., 2020) and SP-gra2seq (Zang et al., 2023b).

However, whilst existing methods (Zang et al., 2021; 2023a; Wang et al., 2022) exhibit effective control on generating sketches with certain global property, they are unable to perform finer control on strokes. For example, researchers have focused on synthesizing sketches of particular categories, such as generating a "cat", but have difficulty in manipulating the shape of certain parts (e.g., the body) of the "cat". Moreover, for users who lack expertise, completing sketches in a single attempt is challenging, and the selected strokes may require multiple modifications. This paper attempts to present a model to mimic human sketch editing at the stroke-level as in Fig. 1.

To achieve the stroke-level editing, it is a key obstacle to pinpoint the strokes that require editing. For the conventional method (Ha & Eck, 2017) using a sequence of points to represent sketches, although the segments determined by the pen states can be directly used as strokes, the lengths of the obtained strokes are not the same, which is not convenient for editing the strokes and updating the sketch sequence. Rasterizing a sketch into an image is a common operation in sketch studies (Chen et al., 2017; Yu et al., 2015; 2016). However, these image-based methods lost details of the drawing order and the way sketch are drawn, making it more difficult to get the stroke information. Recently,

Figure 1: (Arrow left) Original sketches. (Arrow right) Edited sketches generated by our model.

the work (Qu et al., 2023) provided an effective way to break down the sketch sequence into strokes for down stream tasks, where the stroke segments are padded to be of the same length. Inspired by this idea, we develop a stroke encoder to encode each stroke separately, without exchanging information with other stroke. This approach provides the flexibility to select strokes and edit them in the latent space of the encoder while minimizing the impact on the content of the rest part of the sketch.

Another challenge for stroke-level editing is how to appropriately place the strokes after the editing is done. As given in the second row of Fig. 1, if we replace the cat's body with the sheep's body, the cat's head moves from the right to the left side of the image. If the cat's head is still in its original position, the generated sketch will be unrealistic. Here, we develop a diffusion model (Ho et al., 2020) for accurate stroke placement. The diffusion model generates the stroke locations progressively through the denoising process, based on the features of all strokes to be synthesized. The diffusion model extends beyond the generation of single-category sketches, enabling the creation of more diverse results, e.g., a pig with wing-like ears. Furthermore, we fuse the stroke embeddings with the generated stroke locations, and devise a sequence decoder to synthesize the final manipulated sketch. The stroke encoder and the sequence decoder are jointly pre-trained under the autoencoder paradigm, with an extra image decoder to learn the local structure of sketches.

In summary, we propose a novel sketch editing method called **SketchEdit** and our contributions are as follows: (i) We develop the traditional task of sketch synthesis into a more controllable sketch editing task at the stroke-level for the first time. The proposed SketchEdit achieves this purpose well and enables the generation of creative sketches. (ii) We present a fresh perspective on the placement of sketch strokes, where strokes are synthesized akin to assembling building blocks. Given a set of base strokes, we first generate meaningful placements for them, and then combine the strokes into a meaningful sketch. (iii) Experiments show that the our method performs significantly better than the state-of-the-art sketch generation models for the task of sketch reconstruction. This guarantees that the edited sketch effectively retains the visual properties of the original sketch.

## 2 RELATED WORK

**Sketch generation.** Sketching, as a practical communication tool and medium for emotional expression, is impressive and expressive. Its related generative tasks have attracted the interest of researchers (Zhou et al., 2018; Das et al., 2021; Ge et al., 2020). An essential work to this is Sketch-RNN (Ha & Eck, 2017), which is facilitating research into deep learning for the imitation of human drawing. The Sketch-RNN is comprised of a bidirectional Long Short-Term Memory (LSTM) (Hochreiter & Schmidhuber, 1997) encoder and a unidirectional LSTM decoder. Although Sketch-RNN is capable of accurately capturing the connection between drawing points, it falls short in perceiving the local structural information of images. Therefore, the subsequent methods (Chen et al., 2017; Song et al., 2018) convert the sequence of sketches into rasterized images and introduce Convolutional Neural Networks (CNNs) (LeCun et al., 1998) as a replacement or supplement to the LSTM encoder. To improve the representational capabilities of the models, graph neural networks (GNNs) (Scarselli et al., 2008) are introduced on top of the image representation (Su et al., 2020; Qi et al., 2022; 2021; Zang et al., 2023b). These methods construct graphs by temporal proximity, spatial proximity or synonymous proximity. Another method to improve performance is to use a Gaussian Mixture Model (GMM) to model the latent space and incorporate Rival Penalized Competitive Learning (RPCL) (Xu et al., 1993) to automatically select the number of Gaussians (Zang et al., 2021; 2023a). However, as mentioned before, previous sketch generation models have struggled to decouple specific strokes, so the proposed SketchEdit takes strokes as input rather than images or drawing points.

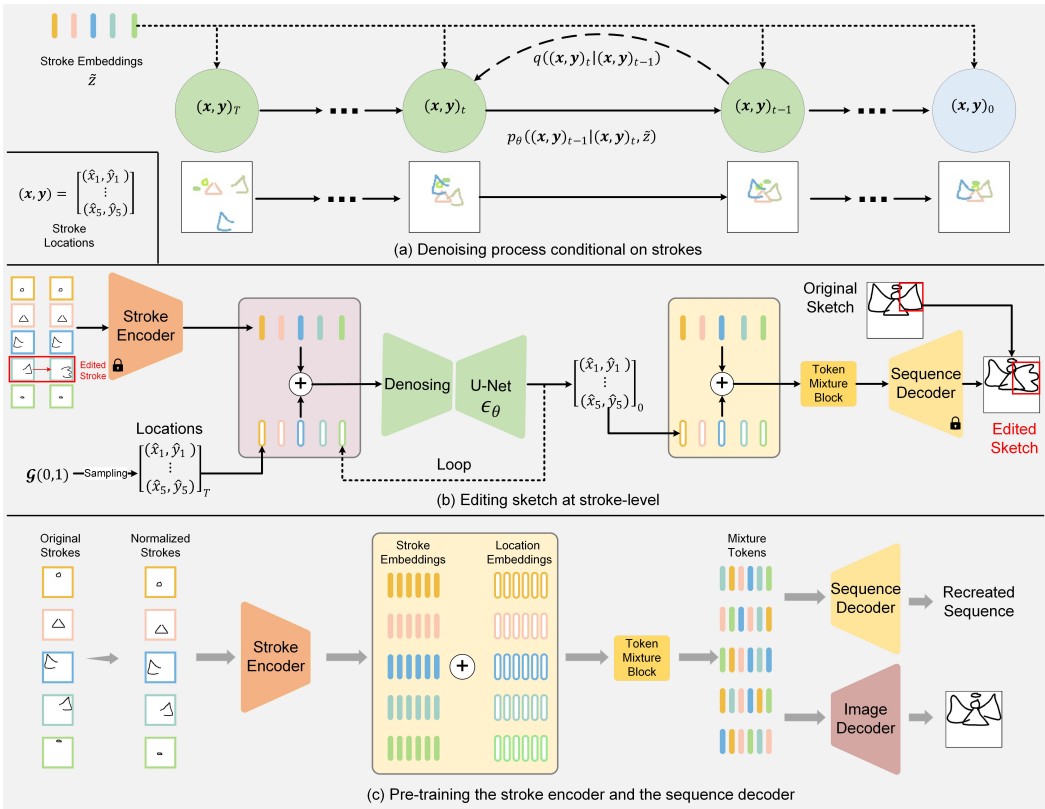

Figure 2: The overview of the proposed SketchEdit. (a) Denoising process conditional on strokes. Essentially, the goal is to reorganize strokes with confusing positions into meaningful sketches. (b) The pipeline for sketch editing by our method. The edited strokes are replaced at the input (or in the latent space) against the target strokes, and then the inverse denoising process is used to obtain meaningful stroke positions from the random noise. (c)Pre-training the stroke encoder and the sequence decoder which are used to generate stroke embeddings and synthesis target sketches for sketch editing task.

**Diffusion models.** Diffusion models (Sohl-Dickstein et al., 2015) have led to a boom in research, particularly in the field of image synthesis (Ho et al., 2020; Dhariwal & Nichol, 2021). Text-to-image (T2I) generation is a widely recognized application of diffusion models, which enables the rapid generation of artwork by providing prompts as a cue to large models, such as DALL·E2 (Ramesh et al., 2021), Imagen (Saharia et al., 2022), GLIDE (Nichol et al., 2021) and stable diffusion (Rombach et al., 2022). However, certain information remains difficult to convey solely through text, leading to the emergence of visual cues as conditions for diffusion models. Sketches are an effective tool for responding to structural information and are therefore regarded as control conditions by PITI (Voynov et al., 2023), ControlNet (Zhang & Agrawala, 2023), UniControl(Qin et al., 2023), T2I-Adapter (Mou et al., 2023), and other methods. Recently some diffusion models (Wang et al., 2022; Das et al., 2023) about sketches have been proposed, which focus on modeling the points of the sketch rather than the stroke locations . Different from the methods mentioned above, where the forward process and reverse denoising process are conducted on images, they consider the points in the sketch sequence as targets. The feasibility of this idea was verified experimentally. Inspired by their research, this paper investigates the potential use of a diffusion model to model stroke locations.

## 3 METHODOLOGY

SketchEdit is constructed based on diffusion model to edit sketches at the stroke-level. The key step is to predict the locations of the strokes. This is achieved by the reverse denoising process

of the diffusion model conditioned on stroke embeddings, as shown in Fig. 2(a). The SketchEdit decouples sketch into several strokes without position information, allowing the user to conveniently select strokes for editing. Strokes and generated locations are eventually fed into a sequence decoder to synthesis the edited sketch. The pipeline of editing sketches are illustrated in Fig. 2(b).

## 3.1 Sketch representation

A sketch is represented by a **sequence** of $L_p$ points, i.e., $\boldsymbol{\tau} = (\boldsymbol{p}_1, \boldsymbol{p}_2, ..., \boldsymbol{p}_{L_p})$. Each point $\boldsymbol{p}_i$ is a vector containing five elements. The first two are the coordinates of the absolute position, while the last three uses the one-hot vector format to represent the three pen states of lift, touch, and the end of sketch. To proceeds in the stroke-level, the sketch sequence is broken down into a series of **strokes**, i.e., $(\boldsymbol{s}_1, \boldsymbol{s}_2, ..., \boldsymbol{s}_{L_s})$, where $L_s$ denotes the number of strokes. We use $(\boldsymbol{x}, \boldsymbol{y}) = [(x_1, y_1), (x_2, y_2), \ldots, (x_{L_s}, y_{L_s})]$ to record the locations of the strokes, which are the coordinates of the first point of the stroke. In this paper, we also define the normalized stroke sequence $\tilde{\boldsymbol{s}}_i$ by subtracting the location $(x_i, y_i)$ of the stroke from the coordinates of all the points in the stroke.

## 3.2 Diffusion model for forecasting locations

**Forward process.** Given a set of stroke locations $(\boldsymbol{x}, \boldsymbol{y})_0 \sim q((\boldsymbol{x}, \boldsymbol{y})_0)$, we apply the Markov diffusion process in DDPMs (Ho et al., 2020) here. The noise sampled from Gaussian distribution is gradually added to $\boldsymbol{x}$ and $\boldsymbol{y}$:

$$
\begin{aligned}
q((\boldsymbol{x}, \boldsymbol{y})_{1:T} | (\boldsymbol{x}, \boldsymbol{y})_0) &= q((\boldsymbol{x}, \boldsymbol{y})_0) \prod_{t=1}^{T} q((\boldsymbol{x}, \boldsymbol{y})_t | (\boldsymbol{x}, \boldsymbol{y})_{t-1}), \\
q((\boldsymbol{x}, \boldsymbol{y})_t | (\boldsymbol{x}, \boldsymbol{y})_{t-1}) &= \mathcal{N}((\boldsymbol{x}, \boldsymbol{y})_t; \sqrt{1 - \beta_t}(\boldsymbol{x}, \boldsymbol{y})_{t-1}, \beta_t \boldsymbol{I}),
\end{aligned}
\tag{1}
$$

where $\beta_t \in (0, 1)$ represents the noise schedule at time $t$.

**Reverse process.** The reverse process aims to recreate the true locations from a Gaussian noise input $(\boldsymbol{x}, \boldsymbol{y})_T$. Similar with the DDPMs (Ho et al., 2020), A U-Net (Ronneberger et al., 2015) like network is utilized to predict the noise $\epsilon_\theta((\boldsymbol{x}, \boldsymbol{y})_t, t)$. However, stroke locations have no explicit semantic information, so it is necessary to introduce strokes as a condition. Thus, the network for predicting noise is modified to $\epsilon_\theta((\boldsymbol{x}, \boldsymbol{y})_t, t, \tilde{\boldsymbol{s}})$. To decrease computational complexity and leverage high-level semantic information, as illustrated in Fig. 2, we utilize the stroke embeddings $\tilde{\boldsymbol{z}}$ as the condition rather than the strokes $\tilde{\boldsymbol{s}}$. The reverse denoising process can be formalized as:

$$
\begin{aligned}
p_\theta((\boldsymbol{x}, \boldsymbol{y})_{t-1} | (\boldsymbol{x}, \boldsymbol{y})_t, \tilde{\boldsymbol{z}}) &= \mathcal{N}((\boldsymbol{x}, \boldsymbol{y})_{t-1}; \boldsymbol{\mu}_\theta((\boldsymbol{x}, \boldsymbol{y})_t, t, \tilde{\boldsymbol{z}}), \boldsymbol{\sigma}_t^2 \boldsymbol{I}), \\
\boldsymbol{\mu}_\theta((\boldsymbol{x}, \boldsymbol{y})_t, t, \tilde{\boldsymbol{z}}) &= \frac{1}{\alpha_t}((\boldsymbol{x}, \boldsymbol{y})_t - \frac{\beta_t}{\sqrt{1 - \bar{\alpha}_t}} \epsilon_\theta((\boldsymbol{x}, \boldsymbol{y})_t, t, \tilde{\boldsymbol{z}})),
\end{aligned}
\tag{2}
$$

where $\alpha_t = 1 - \beta_t$ and $\bar{\alpha}_t = \prod_{i=1}^{t} \alpha_i$. In practice, we use the DDIM-based (Song et al., 2020) generation process for accelerated sampling.

## 3.3 Editing freehand sketches at the stroke-level

In this subsection, we provide the process of editing sketch at the stroke-level. First, pick the to be edited stroke $\tilde{\boldsymbol{s}}_i$ from the sketch $\boldsymbol{\tau}$. The edited stroke $\hat{\boldsymbol{s}}_i$ can either be drawn by the user or selected from the stroke gallery to replace $\tilde{\boldsymbol{s}}_i$. Taking the angle shown in Fig. 2(b) as an example, we have obtained the strokes $\hat{\boldsymbol{s}}(\tilde{\boldsymbol{s}}_1, \tilde{\boldsymbol{s}}_2, \tilde{\boldsymbol{s}}_3, \hat{\boldsymbol{s}}_4, \tilde{\boldsymbol{s}}_5)$ after editing. Then, the stroke encoder calculates the stroke embeddings $\hat{\boldsymbol{z}}(\tilde{\boldsymbol{z}}_1, \tilde{\boldsymbol{z}}_2, \tilde{\boldsymbol{z}}_3, \hat{\boldsymbol{z}}_4, \tilde{\boldsymbol{z}}_5)$. As the encoding process does not involve the exchange of stroke information, stroke substitution in the latent space, such as replacing $\tilde{\boldsymbol{z}}_4$ with $\hat{\boldsymbol{z}}_4$, is also possible.

Next, we apply the reverse process of diffusion model to denoise random noise $(\hat{\boldsymbol{x}}, \hat{\boldsymbol{y}})_T$ conditional on $\hat{\boldsymbol{z}}$, resulting generated stroke locations $(\hat{\boldsymbol{x}}, \hat{\boldsymbol{y}})_0$. Finally, the stroke embeddings $\hat{\boldsymbol{z}}$ and the stroke locations $(\hat{\boldsymbol{x}}, \hat{\boldsymbol{y}})_0$ are fed into the token mixture block and sequence decoder to synthesis the edited sketch $\hat{\boldsymbol{\tau}}$.

### 3.4 Construct the Stroke Encoder and the Sequence Decoder

After converting the sketch sequence to the normalized stroke representation, the resulting tensor $\tilde{s} \in \mathbb{R}^{L_s \times L_n \times 5}$ is obtained, where $L_n$ is the number of points in a stroke. A position-sensitive block must act as the backbone of the stroke encoder to extract features form $\tilde{s}$. Because significant changes in the shape of the stroke occur when any two points in the sequence are interchanged. Token-based MLPs (Tolstikhin et al., 2021) fulfil this requirement, and thus we consider gMLP (Liu et al., 2021) as the basic component. Since we do not wish for any exchange of information to occur during the encoding stage between the strokes, we can intuitively treat the first dimension of $\tilde{s}$ as the batch size.

Several layers are used to extract the stroke embeddings $\tilde{z}$. Firstly, each point in a stroke is treated as a token, which then interacts through the network with other points. Next, these tokens are summed for aggregation to get $\tilde{z}_{enc} \in \mathbb{R}^{L_s \times d_{model1}}$, where $d_{model1}$ denotes the dimension of the tokens. The stroke embeddings $\tilde{z} \in \mathbb{R}^{L_s \times d_{model2}}$ are calculated as followings:

$$\tilde{\mu}, \tilde{\sigma} = f_{linear}(\tilde{z}_{enc}), \tilde{\mu}, \tilde{\sigma} \in \mathbb{R}^{L_s \times d_{model2}},$$
$$\tilde{z} = \tilde{\mu} + \tilde{\sigma} \times \epsilon_{enc}, \epsilon_{enc} \sim \mathcal{G}(\mathbf{0}, \mathbf{I}), \tag{3}$$

where $f_{linear}(\cdot)$ and $d_{model2}$ represents a linear projection and the dimension of stroke embeddings, respectively. The reparameterization trick (Kingma & Welling, 2013) employed in Equation 3 serves to effectively constrain the latent space, resulting in improved continuity.

Then, we map the stroke locations $(\boldsymbol{x}, \boldsymbol{y}) \in \mathbb{R}^{L_s \times 2}$ to the location embeddings $\boldsymbol{z}_{loc} \in \mathbb{R}^{L_s \times d_{model2}}$. The summation of $\tilde{z}$ and $\boldsymbol{z}_{loc}$ is fed into a token mixture block to mixture the information of different strokes. The resulting $\boldsymbol{z}_{mix} \in \mathbb{R}^{L_s \times d_{model2}}$ is subsequently sent to both the sequence decoder and the image decoder. The decoders utilize spatial projection to increase the number of tokens before reconstructing either the sequence $\tilde{\tau}(\tilde{p}_1, \tilde{p}_2, ..., \tilde{p}_{L_p})$ or the image $\tilde{I}$. The backbone of the proposed token mixing block and sequence decoder is gMLP, while the image decoder is built based on CNNs. Thanks to the powerful global capture capability of gMLP, we can decode all sequence points simultaneously, rather than using the autoregressive approach (Ha & Eck, 2017; Chen et al., 2017; Su et al., 2020). This still result in good reconstruction outcomes.

### 3.5 Two-Stage Training

**Pre-train the stroke encoder and the sequence decoder.** After completing end-to-end training, the stroke encoder and the sequence decoder can effectively reconstruct sketches. There are three training objectives. The first is for the output of the sequence decoder, where our goal is to minimize the negative log-likelihood function of the generated probability distribution:

$$\mathcal{L}_{seq} = -\mathbb{E}_{u_\phi(\tilde{z}|\tilde{S})} \log v_\xi(\tilde{\tau}|\tilde{z}, (\boldsymbol{x}, \boldsymbol{y})). \tag{4}$$

The training goal in Sketch-RNN (Ha & Eck, 2017) also pursues this aim, with the variance being the absolute or relative coordinates modeling. For calculating the image reconstruction loss $\mathcal{L}_{img}$, we utilize the traditional mean square error (MSE). To improve the representational power of the model (Zang et al., 2021; 2023a), GMM modeling is carried out in the encoder's latent space. We initialize $K$ Gaussian components and the appropriate number is determined automatically with the aid of RPCL (Xu et al., 1993). The corresponding loss function is formalized as follows:

$$\mathcal{L}_{GMM} = \sum_{i=1}^{L_s} KL(u_\phi(\tilde{z}_i, k|\tilde{s}_i)||o_\psi(\tilde{z}_i, k)), \tag{5}$$

where $\tilde{z}_i$ is the stroke embedding correspond to the stroke $s_i$ and the KL term is calculated as in (Jiang et al., 2016). The parameters of the GMM are learned by an EM-like algorithm, details of which can be found in (Zang et al., 2021). In summary, the overall objective is:

$$\mathcal{L}_{AE} = \mathcal{L}_{seq} + \mathcal{L}_{img} + \lambda \mathcal{L}_{GMM}, \tag{6}$$

where $\lambda$ is a hyperparameter and we set it to 0.0001 in practice.

**Train the diffusion model.** In this stage, the previously trained parameters of the stroke encoder and the sequence decoder are fixed, and the following are the training objectives of the diffusion model:

$$\min_\theta \mathbb{E}||\epsilon - \epsilon_\theta((\boldsymbol{x}, \boldsymbol{y})_t, t, \tilde{z}))||_2^2. \tag{7}$$

## 4 EXPERIMENT

### 4.1 PREPARATION

**Dataset.** Two dataset are selected from the largest sketch dataset QuickDraw (Ha & Eck, 2017) for experiments. **DS1** is a 17-category dataset (Su et al., 2020; Qi et al., 2022). The specific categories are: *airplane, angel, alarm clock, apple, butterfly, belt, bus, cake, cat, clock, eye, fish, pig, sheep, spider, umbrella, the Great Wall of China*. These categories are common in life and the instances in the categories are globally similar in appearance. **DS2** (Zang et al., 2021) is a widely used, comparatively small dataset for synthesized sketches, comprising five categories: *bee, bus, flower, giraffe, and pig*. Each category contains 70000 sketches for training and 2500 sketches for testing.

**Implement Details.** The AdamW optimizer (Loshchilov & Hutter, 2017) is applied to train the proposed model with parameters $\beta_1 = 0.9$, $\beta_2 = 0.999$, $\epsilon = 10^{-8}$ and $weight\ decay = 0.01$. We use the CosineAnnealingLR scheduler (Smith & Topin, 2019) with the peak learning rates are 0.002 and 0.0005 for the pre-trained model and the diffusion model, respectively. All the sketch is padded to the same length, i.e. $L_p = 180$. Each sketch is break down into $L_s = 25$ strokes and each stroke contains 96 points. The method are implemented by pytorch and trained on 5 RTX 2080Ti GPUs. For the pre-trained network, we train it with 15 epochs and the batch size is 200. There are 8 gMLP blocks in the stroke encoder with $d_{model1} = 96$ and $d_{ffn1} = 384$. The token mixture block and the sequence decoder includes 2 and 12 gMLP blocks, respectively. We set $d_{model1} = 128$ and $d_{ffn1} = 512$ for these blocks. Drop path rate is set to 0.1. We train the U-Net of the diffusion model with 40 epochs with the batch size is 768. The encoder and the decoder both consist of 12 gMLP blocks with the drop path rate is 0.1. The $d_{model}$ and $d_{ffn}$ in these blocks are 96 and 384, respectively. For the forward process and the reverse denoising process, we set the time step $T = 1000$. We consider the linear noise schedule for the model with $\beta_1 = 0.0001$ and $\beta_T = 0.02$. We take 60 steps for DDIM sampling in default.

**Competitors.** We consider 3 types of models as the competitors for sketch reconstruction. Sketch-RNN (Ha & Eck, 2017) employs a VAE (Kingma & Welling, 2013) framework to learn sketch representations from **sequences**. Sketch-pix2seq (Chen et al., 2017) takes sketch **images** as input to learn local structural information form sketches. RPCL-pix2seq (Zang et al., 2021) develops the decoder of Sketch-pix2seq into a dual-branch architecture and constrain the code with GMM. Based on the rasterized sketch images, SketchHealer (Su et al., 2020), SketchLattice (Qi et al., 2021), and SP-gra2seq (Zang et al., 2023b) introduce the GNNs for better representations. The **graphs** are constructed based on time, position, and synonymous proximity, respectively.

**Metrics.** To evaluate the performance of the SketchEdit, we select Rec (Zang et al., 2021), FID (Heusel et al., 2017), LPIPS(Zhang et al., 2018), and CLIP Score (Radford et al., 2021; Hessel et al., 2021) as the metrics. To classify whether the recreated sketches are belongs to the original category, two sketch-a-nets (Yu et al., 2015) are trained on DS1 and DS2, respectively. Rec is the success rate of recognition.

### 4.2 EDITING SKETCHES AT THE STROKE-LEVEL.

Stroke-level sketch editing involves modifying distinct strokes while minimizing the impact on the overall structure. Sketches typically consist of various basic shapes, and strokes from other sketches can be conveniently reused to edit the intended sketch, as illustrated in Fig. 3. The recycled shapes may comprise constituents from the identical class with clearly defined meanings, for example, an airplane fuselage, an umbrella handle, and so on. Using interpolation techniques to generate additional components with uniform semantics can efficiently produce a substantial amount of novel sketches. Apart from that, creative editing enables a sensible synthesis of strokes from different categories of sketches. Some examples are provided in Fig. 3, for instance, the alarm clock's bells have been replaced by apple stems, and the SketchEdit has found a "logical" place for the apple stem.

Although our method is flexible when it comes to editing strokes, identifying appropriate metrics for evaluation remains challenging. Therefore, we employ an intermediate task for evaluation. Initially, we utilize the diffusion model for position locations based on normalized strokes. Then sketches are synthesised with the generated locations. Subsequently. Table 1 reports the experimental results for this intermediate task. Compared to the performance of recreating sketches with the original loca-

Table 1: The performance for recreating sketches from normalized strokes with unknown locations. Diffusion models are involved in the prediction of locations. SketchEdit(o_l) denotes recreating sketches with the original locations but not the generated locations.

| Model | DS1 | | | | DS2 | | | |
|---|---|---|---|---|---|---|---|---|
| | Rec(↑) | FID(↓) | LPIPS(↓) | CLIP-S(↑) | Rec(↑) | FID(↓) | LPIPS(↓) | CLIP-S(↑) |
| SketchEdit | 78.79% | 3.79 | 0.29 | 94.59 | 85.89% | 7.77 | 0.37 | 91.96 |
| SketchEdit(o_l) | 84.32% | 3.12 | 0.11 | 96.73 | 93.42% | 5.88 | 0.19 | 94.25 |

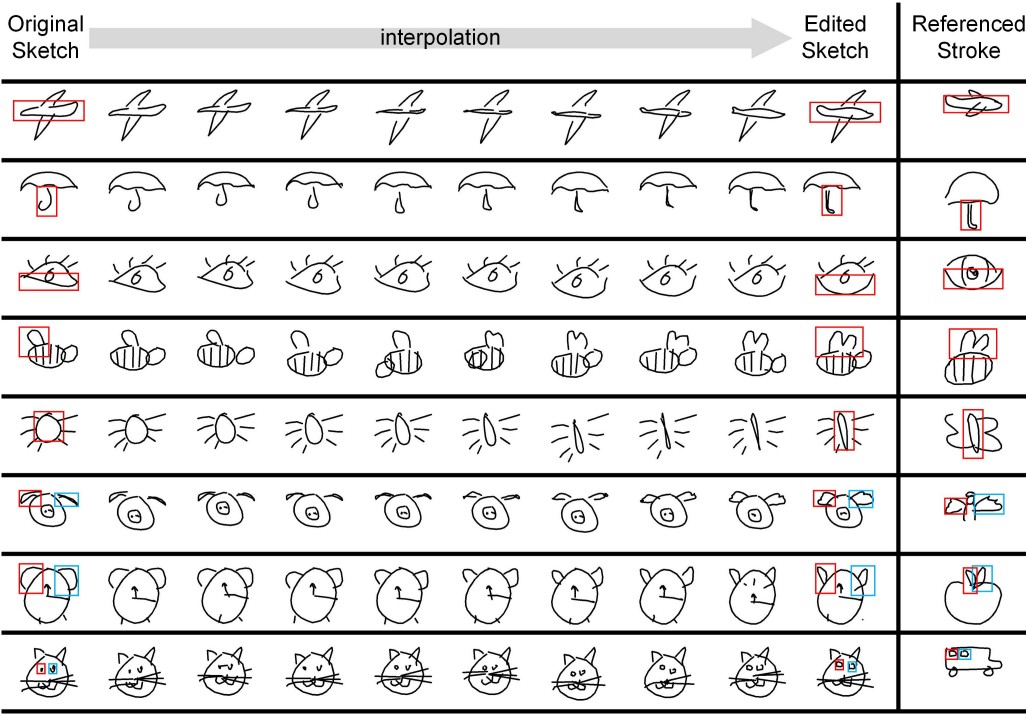

Figure 3: Exemplary sketch editing results. Boxes of the same color in each row denote the respective modified strokes. Creative sketches can be generated through the interpolation between strokes in the latent space and the locations of strokes are produced by our diffusion model.

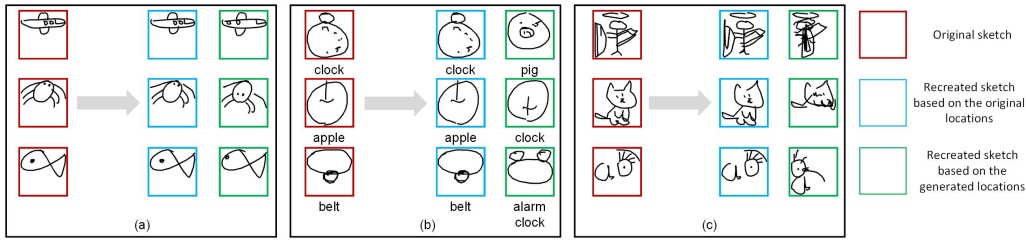

Figure 4: Examples of issues caused by the utilization of generated locations in sketch reconstruction. (a) Some of the components have moved. (b) The sketch category changes. (c) The results synthesised are meaningless.

tions, the results with the generated ones experience a significant decrease , especially in the LPIPS metric. There are three primary factors contributing to the decline in semantic similarity between the recreated sketches and their corresponding sketches, as shown in Fig. 4. Reasonable movement of components and generation of different classes of sketches is tolerated because our diffusion model does not provide more guidance just to get appropriate stroke positions. An important reason for the

generation of meaningless sketches is that the target sketches are rarer patterns, and our diffusion model struggles to accurately predict stroke locations against them.

## 4.3 COMPARISON WITH STATE-OF-THE-ARTS FOR SKETCH RECONSTRUCTION

Sketch reconstitution requires the model to recreate the sketch $\tilde{\tau}$ from the input $\tau$. High-quality sketch reconstruction is essential to maintaining a consistent visual appearance between the edited sketch and the original sketch. In this subsection, we compare the SketchEdit with other sketch synthesis methods. For a fair comparison, our model uses the original stroke Locations instead of the generated ones.

**Qualitative analysis.** Table 2 reports the sketch reconstruction performance of the proposed method and its competitors. Our model significantly outperforms other methods across all metrics. The SketchEdit model captures global dependencies in sketch sequences more efficiently, while the proposed sequence decoder addresses the challenge of stacked layers in LSTM and the deeper network improves reconstruction results. However, due to the data-driven nature of the gMLP block, it lacks adequate inductive bias, resulting in a less prominent advantage of SketchEdit on the smaller DS2 compared to DS 1. For Sketch-RNN (Ha & Eck, 2017), the FID metrics and other metrics present a distinct phenomenon. The inputs for the Sketch-RNN and the SketchEdit consist of sketch sequences or strokes, without requiring the sketches to be rasterized into images. There exists a considerable domain gap between the sequences and the images, resulting in a disparity between the distributions learned by the image-based approach and the sequence sketches.

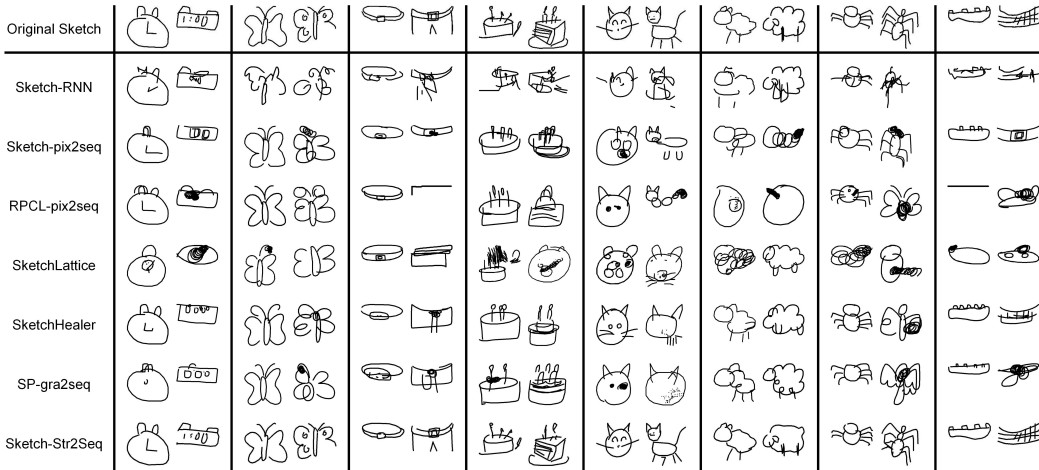

Figure 5: The exemplary result of reconstructed sketches by the proposed SketchEdit and other models. The categories from left to right are alarm clock, butterfly, belt, cake, cat, sheep, spider and the Great Wall of China.

Table 2: The performance for sketch reconstruction with original stroke locations.

| Model | DS1 | | | | DS2 | | | |
|---|---|---|---|---|---|---|---|---|
| | Rec(↑) | FID(↓) | LPIPS(↓) | CLIP-S(↑) | Rec(↑) | FID(↓) | LPIPS(↓) | CLIP-S(↑) |
| Sketch-RNN | 64.51% | 6.87 | 0.33 | 91.82 | 77.74% | 10.45 | 0.40 | 90.29 |
| Sketch-pix2seq | 66.99% | 42.03 | 0.34 | 90.04 | 88.36% | 42.78 | 0.37 | 90.22 |
| RPCL-pix2seq | 69.86% | 44.09 | 0.32 | 90.37 | 90.66% | 27.32 | 0.35 | 90.80 |
| SketchLattice | 48.88% | 48.70 | 0.44 | 87.06 | 77.54% | 50.92 | 0.45 | 87.80 |
| SketchHealer | 76.76% | 21.62 | 0.32 | 92.15 | 90.93% | 24.43 | 0.36 | 91.28 |
| SP-gra2seq | 76.60% | 21.92 | 0.33 | 92.01 | 91.12% | 21.69 | 0.37 | 91.15 |
| SketchEdit | **84.32**% | **3.12** | **0.11** | **96.73** | **93.42**% | **5.88** | **0.19** | **94.25** |

**Quantitative analysis.** Fig. 5 presents the qualitative comparisons. Compared to other approaches, SketchEdit is capable of reconstructing sketches with high-quality, without introducing additional noisy strokes, while preserving the structural patterns of the sketches. To prevent generated sketches from changing category, the model must first learn an accurate representation of the category-level.

A failure case is that Sketch-pix2seq reconstructs the last column of the Great Wall into a belt. Capturing structural information at the instance-level is a challenging undertaking. While nearly all the competitors reproduced "cakes" as "cakes", the generated results displayed significant structural changes. Furthermore, the existence of multiple styles within the same sketch category poses a challenge to sketch reconstruction. The proposed SketchEdit shows significant preservation of detail about sketch instances, which is the basis for our sketch editing task.

## 4.4 ABLATION STUDY

In this subsection, we discuss the effectiveness of the image decoder and the token mixture block. We conduct the ablation study on DS1. SketchEdit(wo_i) and SketchEdit(wo_s) denote that no image decoder is included and no token mixture block is shared between the two decoders, respectively.

Table 3: The performance for sketch reconstruction with the original locations and the generated locations.

| Model | Original Locations | | | | Generated Locations | | | |
|---|---|---|---|---|---|---|---|---|
| | Rec($\uparrow$) | FID($\downarrow$) | LPIPS($\downarrow$) | CLIP-S($\uparrow$) | Rec($\uparrow$) | FID($\downarrow$) | LPIPS($\downarrow$) | CLIP-S($\uparrow$) |
| SketchEdit(wo_i) | 83.56% | 3.80 | 0.13 | 96.34 | 78.19% | 4.45 | 0.30 | 94.36 |
| SketchEdit(wo_s) | 84.20% | 3.21 | 0.12 | 96.45 | 78.41% | 3.93 | 0.29 | 94.42 |
| SketchEdit(full) | **84.32%** | **3.12** | **0.11** | **96.73** | **78.79%** | **3.79** | **0.29** | **94.59** |

Table 3 reports the results of the ablation experiments. SketchEdit(full) and SketchEdit(wo_s) with image encoders have performance advantages over Str2Seq(wo_c). This is because the use of image reconstruction allows the network to learn shape information and spatial relationships. Similarly, SketchEdit(wo_s) would make learning image-related information difficult for the token mixture block at the sequence decoder. As shown in Fig. 6, some strokes overlap in the results produced by SketchEdit(wo_s) and SketchEdit(wo_c) which reduces the quality of the recreated sketch. In addition, SketchEdit(full) has marginally fewer parameters compared to SketchEdit(wo_s) as it only employs a single token mixture block.

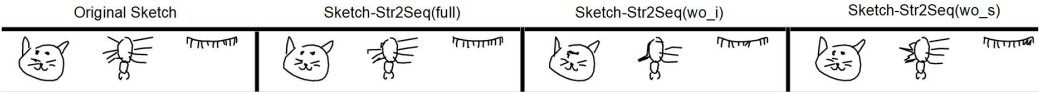

Figure 6: Comparison of recreated sketches across various models in ablation studies.

## 5 CONCLUSION AND FUTURE WORK

In this paper, we develop the traditional sketch synthesis task to the more controllable sketch editing task at the stroke-level and propose the SketchEdit to realize it. We have focused on decoupling independent strokes from sketches to enable editing operations at the stroke-level. The core of our methodology is to employ the diffusion model to acquire reasonable positions and recreate meaningful sketches based on the strokes. Experimental results demonstrate that SketchEdit can edit sketches without altering categories and facilitate the production of innovative sketches across various categories. Meanwhile, SketchEdit which efficiently preserves the spatial structure of sketches and supports the parallel reconstruction of sketch sequences, surpasses the state-of-the-art methods significantly in the sketch reconstruction task.

While our work contributes to the research on the controllability of sketch generation, there remain a number of issues that require further improvement in the future. (i) One aspect is more flexible control, e.g., given no reference strokes, the model is able to automatically obtain a large number of reasonable strokes to replace the ones to be edited. (ii) Although our technique is capable of producing high-quality outcomes, certain strokes are subject to over-smoothing, resulting in dissimilarities from human drawing styles. Therefore, it is worthwhile further exploring the design of models that align with human drawing styles and efficiently generate sequences. (iii) The design of metrics also a tricky issue. Most of the existing metrics for measuring the results of image generation are based on natural images rather than abstract sketches. Thus, the development of novel sketch evaluation metrics for recreated or edited sketches also constitute a key aspect of forthcoming research.

# 6 REPRODUCIBILITY STATEMENT

To ensure the reproducibility of the proposed methodology, the details of each module are provided in the Appendix. The full project code and the the detailed usage are provided in the supplementary material.

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

## A  THE DETAILS OF THE PRE-TRAINED MODEL

In the main text, we have given the details of the encoder and token mixture block setup, next we will provide the details of the decoders. Fig. 7 shows the architecture of some components in our AE. The token mixture block gives the embedding $\tilde{z}_{mix} \in \mathbb{R}^{L_s \times d_{model2}}$ after mixing the independent strokes by gMLP blocks Liu et al. (2021). To decoder sequence or image from $\tilde{z}_{mix}$, expanding the number of tokens is necessary and Fig. 7 (a) is the structure of the component. Several convolutional layers and deconvolutional layer are used in the image decoder to upsampling the feature map, as shown in Fig. 7 (c). For sequence decoder, the recreated sequences are not predicted directly, instead the parameters $\boldsymbol{o}(\boldsymbol{o}_1, \boldsymbol{o}_2, ..., \boldsymbol{o}_{L_p})$ of mixture density model (MDN) are output:

$$
\begin{aligned}
&\boldsymbol{o}_i = W_o h_i + b_o, \boldsymbol{o}_i \in \mathbb{R}^{6M+3}, \\
&[(\hat{\Pi}_1, \mu_x, \mu_y, \hat{\sigma}_x, \hat{\sigma}_y, \delta_x, \delta_y, \hat{\rho}_{xy})_1, ..., (\hat{\Pi}_1, \mu_x, \mu_y, \hat{\sigma}_x, \hat{\sigma}_y, \delta_x, \delta_y, \hat{\rho}_{xy})_M, (\hat{q}_1, \hat{q}_2, \hat{q}_3)] = \boldsymbol{o}_i, \\
&\sigma_x = exp(\hat{\sigma}_x), \sigma_y = exp(\hat{\sigma}_y), \rho_{xy} = tanh(\hat{\rho}_{xy}), \\
&\Pi_k = \frac{\hat{\Pi}_k}{\sum_{j=1}^M \hat{\Pi}_j}.
\end{aligned}
\tag{8}
$$

The first $M$ sets of $\boldsymbol{o}_i$ are the parameters for a GMM with $M$ normal distributions, which is used to model the ordination $(x, y)$. Hence $(x, y)$ can be expressed in probabilistic form as follows:

$$
p(x, y) = \sum_{j=1}^M \Pi_j \mathcal{N}(x, y | \mu_{x,j}, \mu_{y,j}, \sigma_{x,j}, \sigma_{y,j}, \rho_{xy,j}), \sum_{j=1}^M \Pi_j = 1,
\tag{9}
$$

The last three parameters of $o_i$ are used to model the pen state distributions:

$$
q_k = \frac{\hat{q}_k}{\sum_{j=1}^3 \hat{q}_j}.
\tag{10}
$$

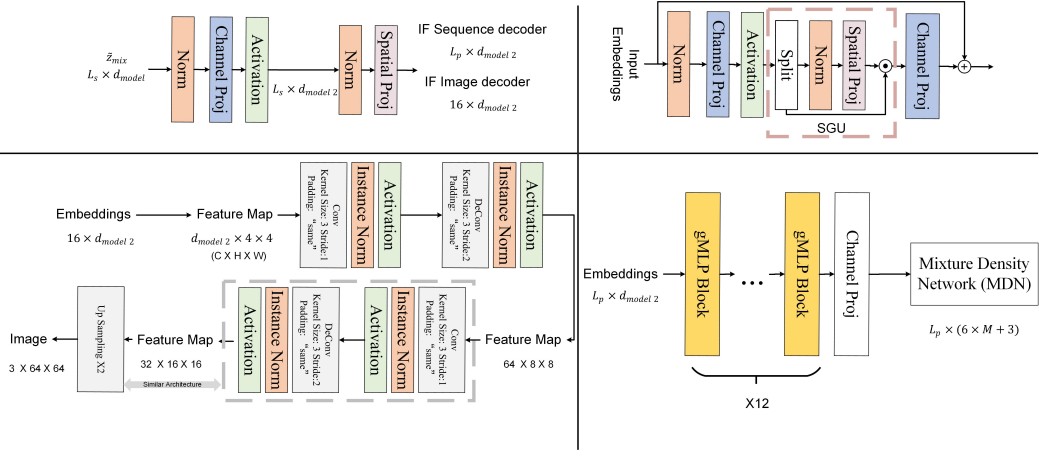

Figure 7: (a) Expanding the number of tokens. (b) gMLP Liu et al. (2021) blocks. (c) The image decoder of our methods. (d) The proposed sequence decoder. Spatial projection and channel projection in the figure are both linear projection.

## B  THE ARCHITECTURE OF U-NET

As shown in Fig. 8 the U-Net (Ronneberger et al., 2015) in the diffusion model is also based on gMLP block. The time step encoding is first initialized using sine-cosine encoding. The final code is then obtained by going through a nonlinear mapping layer and a linear mapping layer in turn.

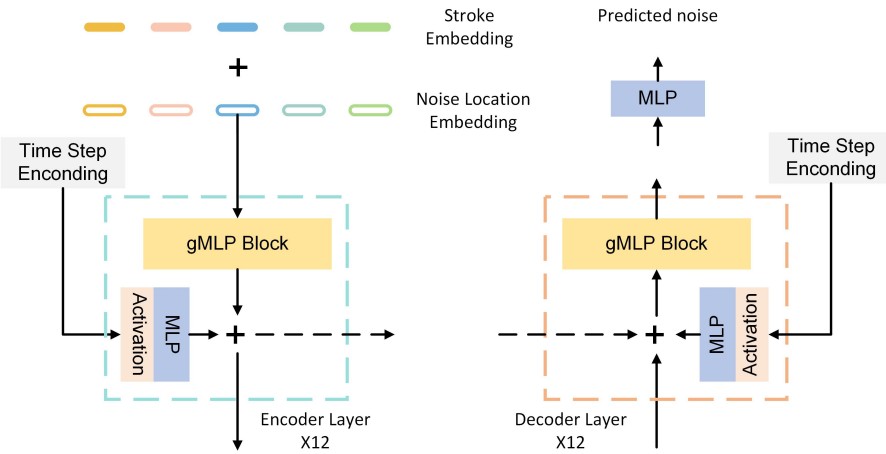

Figure 8: The designed U-shape like network for our diffusion model. A linear projection is included for dimensional changes to the Stroke embeddings output by the encoder.

## C    RECREATE MULTI-STYLE SKETCHES

Fig. 9 shows some examples of SketchEdit's effective reconstruction of multi-style sketches. Benefiting from the strong reconstruction capability for multi-style sketches, the model also maintains the sketch structure features well during the sketch editing task.

Figure 9: Samples of multi-style sketches in the same category. (Left) The original sketches from dataset and (Right) the recreated sketches by the original locations. The categories from top to down are clock, bus, flower, giraffe, pig, fish, apple and bee.

## D    THE DETAILS OF LOSS FUNCTION

In this section, we describe the objectives $\mathcal{L}_{seq}$ and $\mathcal{L}_{GMM}$. $\mathcal{L}_{seq}$ Ha & Eck (2017) is consist of two terms, e.g., maximize the likelihood of the reconstructed distribution and the classification of the pen state. Details are as follows:

$$\mathcal{L}_s = -\frac{1}{L_p} \sum_{i=1}^{N_s} \log \sum_{j=1}^{M} \Pi_j \mathcal{N}(x, y | \mu_{x,j}, \mu_{y,j}, \sigma_{x,j}, \sigma_{y,j}, \rho_{xy,j}),$$

$$\mathcal{L}_p = -\frac{1}{L_p} \sum_{i=1}^{L_p} \sum_{k=1}^{3} p_{k,i} \log \hat{q}_i,$$

(11)

where $N_s$ is the number of points in the original sequence.

For the GMM used to constraint the latent space, we initialize $K = 40$ Gaussian components and utilize a EM-like learning algorithm Zang et al. (2021) to update the parameters of GMM in each training step:

$$E - step : u_\phi^{(t)}(k|\tilde{z}_i) = \frac{o_\psi^{(t-1)}(k)o_\psi^{(t-1)}(\tilde{z}_i|k)}{\sum_j o_\psi^{(t-1)}(j)o_\psi^{(t-1)}(\tilde{z}_i|j)},$$

$$\hat{u}_{ik}^{(t)} = \begin{cases} 1, & winer, k = k^*, k^* = argmax_k u_\phi^{(t)}(k|\tilde{z}_i); \\ -0.0001, & rival, k = u, u = argmax_{k \neq k^*} u_\phi^{(t)}(k|\tilde{z}_i); \\ 0, & otherwise., \end{cases}$$

$$M - step : \hat{\boldsymbol{\mu}}_k^{(t)} = \frac{\sum_i \tilde{z}_i \cdot \hat{u}_{ik}^{(t)}}{\sum_i \hat{u}_{ik}^{(t)}},$$

$$diag(\hat{\boldsymbol{\sigma}}_k^{2(t)}) = \frac{\sum_i \hat{u}_{ik}^{(t)}[(\tilde{z}_i - \boldsymbol{\mu}_k^{(t)})(\tilde{z}_i - \boldsymbol{\mu}_k^{(t)})^T + diag(\tilde{\boldsymbol{\sigma}}_i^2)]}{\sum_i \hat{u}_{ik}^{(t)}},$$ (12)

$$\hat{\alpha}_k^{(t)} = \frac{\sum_{i=1}^{L_s} \hat{u}_{ik}^{(t)}}{L_s}$$

$$Update : \boldsymbol{\mu}_k^{(t)} = (1 - \eta)\boldsymbol{\mu}_k^{(t-1)} + \eta\hat{\boldsymbol{\mu}}_k^{(t)}$$

$$\boldsymbol{\sigma}_k^{2(t)} = max\{(1 - \eta)\boldsymbol{\sigma}_k^{2(t-1)} + \eta\hat{\boldsymbol{\sigma}}_k^{2(t)}, 10^{-5}\boldsymbol{I}\},$$

$$\alpha_k^{(t)} = max\{(1 - \eta)\alpha_k^{(t-1)} + \eta\hat{\alpha}_k^{(t)}, 0\},$$

where $\hat{\alpha}_k$, $\boldsymbol{\mu}_k$ and $diag(\boldsymbol{\sigma}_k^2)$ denote the mixing probability, Gaussian centroid and diagonal covariance matrix of the k-th Gaussian. And the details about the calculation of $L_{GMM}$, please refer to (Jiang et al., 2016).

