# OpenReview forum: "SketchEdit: Editing Freehand Sketches At The Stroke-Level"
_ICLR.cc/2024/Conference — ICLR 2024 Conference Withdrawn Submission_

### Official Review · Reviewer_zvBb · 2023-10-31

**Soundness:** 3 good
**Presentation:** 2 fair
**Contribution:** 2 fair
**Rating:** 5
**Confidence:** 5

**Summary:**

This work aims to propose a method for stroke-level sketch editing. It addresses the challenges of decoupling strokes from sketches and accurately positioning edited strokes. The proposed approach divides the drawing sequence into individual strokes and learns stroke embeddings using a stroke encoder. Users can easily select and edit strokes at any location. The diffusion process is used to generate precise locations for the strokes based on their features. These stroke embeddings and locations are then fed into a sequence decoder to synthesize the manipulated sketch.

**Strengths:**

- It introduces the idea of stroke-level editing, allowing users to modify specific strokes in a sketch while preserving the overall structure. This is an interesting point, which is meaningful when finer control over stroke is required.

- Another contribution of this work is that the authors employ a diffusion model for stroke placement, i.e., generating stroke locations based on their features. The deployment of diffusion models on stroke placement seems novel.

**Weaknesses:**

- The focus of this work is on sketch editing, however, there are no specific experiments conducted to demonstrate the usefulness of the model. Perhaps the quality of the experiments could be improved by introducing editing-related tasks, such as stroke replacement/modification/interpolation, and providing analysis on how these change or improve the visual outcomes and the obtained matrics scores.

- The novelty of the paper is somehow limited. The proposed method for generating sketches is mainly based on combining existing techniques. Diffusion models have been leveraged to model locations of stroke points [A, B]. The concept of generating parts/strokes and subsequently assembling them to create a sketch is not new either [C]. Please refer to the attached papers for more information.

        [A] ChiroDiff: Modelling chirographic data with Diffusion Models, ICLR 2023
        [B] SketchKnitter: Vectorized Sketch Generation with Diffusion Models, ICLR 2023
        [C] Creative Sketch Generation, ICLR 2020

- From Table 1, compared with using the original locations, the generation results with the generated stroke locations will decrease dramatically, caused by undesired stroke location shift, category changes, and messy stroke placement. Although the authors had provided some analysis on this, this could be a significant limitation of the work.

- Regarding the sketch reconstruction comparison mentioned in section 4.3, the proposed method utilizes the original stroke location rather than the generated ones. It would be beneficial to evaluate the performance of using the full model in conjunction with location generation. This will help readers to understand the proposed method better.

**Questions:**

Please find my major concerns in the weaknesses. Additionally, I have a few questions that require clarification from the authors.

- From Figure 2, it is evident that when a stroke is edited, the remaining strokes remain unchanged. The question arises, how does the decoder accomplish this?
- Will the number of strokes remain unchanged when implementing the proposed model?
- Why stroke normalization is required?

---

> ### Author Response · Authors · 2023-11-17
> **Thanks for your comment!**
>
> We sincerely appreciate your many valuable comments.
>
>
> **Q1: No specific experiments conducted to demonstrate the usefulness of the model.**
>
>
> A1: We will enrich the experiment and subdivide it into subsections on stroke interpolation, stroke replacement, and stroke addition in subsequent revisions of the manuscript, and give a higher-quality qualitative analysis.
>
>
> **Q2: The novelty of the paper is somehow limited.  The proposed method for generating sketches is mainly based on combining existing techniques.**
>
>
> A2: Although there are some similarities between previous techniques and our approach, there are still obvious differences. Our approach differs from [A] and [B] in that they are pure diffusion models that lack a semantic latent space, whereas our approach utilizes the latent space learned from the AE paradigm to be able to edit the sketches more flexibly. For model training in [C], the parts of the sketches need to have explicit labeling information, and these parts are with explicit semantic information, e.g., bird heads. For sketch data, obtaining part labels is costly. Our approach does not require manual labelling and focuses on more basic and simple strokes that are easily accessible and highly reusable.
>
>
> **Q3: The generation results with the generated stroke locations decrease dramatically, which this could be a significant limitation of the work.**
>
>
> A3: Composing a number of abstract, simple strokes that lack positional information into a sensible sketch is extremely challenging. Therefore, it is inevitable that the results of sketches reconstructed by generating locations are not as good as the original locations. Although there is a drop in metrics, the results of our method lacking location information still outperform the baseline model overall.
>
>
> **Q4: It would be beneficial to evaluate the performance of using the full model in conjunction with location generation.**
>
>
> A4: Thank you for your advice. We will merge the two tables for easier reading.
>
>
> **Q5: From Figure 2, it is evident that when a stroke is edited, the remaining strokes remain unchanged. The question arises, how does the decoder accomplish this?**
>
>
> A5: This is due to the separation of stroke position and stroke structure in our encoding stage. The stroke representation extracted by the stroke encoder has high-level semantics and the interference of positional information with shape coding is avoided, thus helping the sequence decoder to efficiently reconstruct the strokes based on the positional and semantic information.
>
>
> **Q6: Will the number of strokes remain unchanged when implementing the proposed model?**
> A6: We counted the number of strokes in the reconstructed sketches in dataset 1 and found that the number of strokes did not change in 36,928 (86.89\%) sketches out of 42,500 results.
>
>
> **Q7: Why stroke normalization is required?**
>
>
> A7: There are two main reasons for this:
> 1. We would like to be able to directly utilize a large number of strokes from the QuickDraw dataset, whereas the same strokes may exist at different locations in different sketches.
> 2. If the original strokes are encoded directly, the additional positional information that would be introduced does not allow us to effectively learn the semantics of the stroke structure.
>
>
> [A] Das, A., Yang, Y., Hospedales, T. M., Xiang, T., \& Song, Y. Z. (2022, September). ChiroDiff: Modelling chirographic data with Diffusion Models. In The Eleventh International Conference on Learning Representations.
>
>
> [B] Wang, Q., Deng, H., Qi, Y., Li, D., \& Song, Y. Z. (2022, September). SketchKnitter: Vectorized Sketch Generation with Diffusion Models. In The Eleventh International Conference on Learning Representations.
>
>
> [C] Ge, S., Goswami, V., Zitnick, L., \& Parikh, D. (2020, October). Creative Sketch Generation. In International Conference on Learning Representations.

---

### Official Review · Reviewer_ihan · 2023-10-31

**Soundness:** 2 fair
**Presentation:** 1 poor
**Contribution:** 2 fair
**Rating:** 3
**Confidence:** 4

**Summary:**

The paper proposes a method to do conditional sketch generation with conditioning done on a stroke level: given a sketch, one or more strokes are replaced, and the diffusion model is used to find the position of the modified strokes to create a plausible sketch. Diffusion model operates on the stroke representation in latent space. To obtain the latent space, an autoencoder is trained with per-stroke encoder and sequence / image decoder. Method is evaluated against other methods by the quality of the sketch reconstruction. The proposed approach allows combining pieces of multiple sketches together, on a stroke level.

**Strengths:**

Originality: The paper proposes using diffusion models for generating sketches at stroke level. This is novel in a limited way (there have been previous works on sketch image generation with diffusion models, mentioned by authors; there have been previous works on handwriting strokes generation with diffusion models, ex. "Diffusion models for Handwriting Generation", Luhman & Luhman 2020).

Quality: The paper provides a detailed description of the approach, likely fostering reproducibility.

Significance: The proposed approach seems to learn a good sketch embedding space, as evident from the interpolations, and recognition quality of the reconstruction.

**Weaknesses:**

Significance: The main contribution of the approach is not articulated well - what are the circumstances in which the specific process described in the paper would be relevant (removing a stroke from a sketch, replacing it with a different stroke, and finding the best positioning for the stokes).

Quality: The comparison on the reconstruction quality misses comparisons to numerous recent works, ex. to name a few
- Abstracting Sketches through Simple Primitives, https://arxiv.org/pdf/2207.13543.pdf - studies the reconstuction quality and breakdown of sketches into individual elements, shows reconstuction quality numbers higher than those shown by the authors (although the comparison is not fair as it evaluates on a full set rather than two subsets)
- Multi-Graph Transformer for Free-Hand Sketch Recognition, https://arxiv.org/pdf/1912.11258.pdf
- Sketchformer: Transformer-based Representation for Sketched Structure, https://arxiv.org/pdf/2002.10381.pdf
- Painter: Teaching Auto-regressive Language Models to Draw Sketches, https://arxiv.org/pdf/2308.08520.pdf

Additionally, the ablation study looks at two small architectural changes (having image decoder, and having a joint token mixture for two decoders), but doesn't really highlight the important questions such as the choice of using the diffusion model, the separate encoding of each stroke compared to the sequence-level embedding, etc.

Finally, the main reported metric, sketch reconstruction quality, has little correlation with the main idea of the paper, namely stroke level editing, and I believe a human study could be a better fit in these scenarios.

Clarity: The writing is often not clear and contains many typos, ex.
- Sec. 3.3, first line: "pick the to be edited stroke"
- Sec 3.3, second paragraph: "resulting [in] generated stroke locations"
- Sec. 3.3, second paragraph: "synthesis the edited sketch"
- Sec. 3.4, 3rd line: ". Because significant" --> "because significant"
- Sec. 3.5, after eq.4: "with the variance being" -> "with the difference being"
- Sec. 4, first line: "two dataset" -> "two datasets"
- Sec. 4, subsection titles: "implement details" --> "implementation details", "competitors" --> "baselines"
- Sec. 4, metrics section: the metric called "Rec" is simply "Accuracy" and should be referred to as such.

**Questions:**

I am most interested in authors pinpointing the exact usecase for their proposed solution, and the metrics that could capture the performance in the suggested usecase.

---

> ### Author Response · Authors · 2023-11-17
> **Thanks for your comment!**
>
> Thank you for your feedback.
>
>
> **Q1:  what are the circumstances in which the specific process described in the paper would be relevant?**
>
>
> A1: For example, our approach is suitable for inspirational education of young children. Sketches are characterized by simplicity and impressiveness. The substitution of different basic strokes into different categories of sketches may develop a child's imagination and migration skills. In addition, the various sketches produced by interpolating between strokes can be used to develop creative skills.
>
>
> **Q2: The comparison on the reconstruction quality misses comparisons to numerous recent works.**
>
>
> A2: We thank the reviewer for the references provided. Below we explain why these papers are not included in the comparison. First, there is a difference in the purpose of the models. [A] and [B] are not methods used for sketch generation. The purpose of [A] is to replace each stroke in the sketch with simple primitives, allowing for a more "regular" representation of the sketch, which is further applied to the abstraction of the sketch. [B] is a sketch recognition model used to classify rather than generate sketches. Proposed in 2020, [C] is a multi-tasking system with sketch reconstruction capabilities. However, it introduces additional labeling during training rather than unsupervised or self-supervised learning, so representative sketch generation work [E], [F], [G] after 2020 has not been compared to it. [D] is an interesting work linking sketch generation to large language models, but its input requires additional prompt. Certainly, in subsequent editions of the manuscript, these articles will be described in RELATED WORK to help readers better understand the progress of the sketch study.
>
>
> **Q3: The ablation study and the clarity of writing.**
>
>
> A3: Thank the reviewer for pointing out the deficiencies in our ablation experiment and writing. In subsequent versions, we will add ablation experiments, including the reasons for choosing the diffusion model. The writing of the manuscript will also be revised to be more smooth.
>
>
> **Q4: The exact usecase for the proposed solution, and the metrics that could capture the performance in the suggested usecase.**
>
>
> A4: As described in A1, our approach applies to the inspirational education of young children. The reason we use the reconstruction task to measure the performance of the method is that previous methods to achieve sketch editing suffer from a tricky problem: the reconstructed sketches are significantly different from the original sketches. The severe changes in the category level and the instance level that occur during sketch reconstruction make the implementation of editing a huge challenge.
>
>
> [A] Alaniz, S., Mancini, M., Dutta, A., Marcos, D., \& Akata, Z. (2022, October). Abstracting sketches through simple primitives. In European Conference on Computer Vision (pp. 396-412). Cham: Springer Nature Switzerland.
>
>
> [B] Xu, P., Joshi, C. K., \& Bresson, X. (2021). Multigraph transformer for free-hand sketch recognition. IEEE Transactions on Neural Networks and Learning Systems, 33(10), 5150-5161.
>
>
> [C] Ribeiro, L. S. F., Bui, T., Collomosse, J., \& Ponti, M. (2020). Sketchformer: Transformer-based representation for sketched structure. In Proceedings of the IEEE/CVF conference on computer vision and pattern recognition (pp. 14153-14162).
>
>
> [D] Pourreza, R., Bhattacharyya, A., Panchal, S., Lee, M., Madan, P., \& Memisevic, R. (2023). Painter: Teaching Auto-regressive Language Models to Draw Sketches. In Proceedings of the IEEE/CVF International Conference on Computer Vision (pp. 305-314).
>
>
> [E] Qi, Y., Su, G., Wang, Q., Yang, J., Pang, K., \& Song, Y. Z. (2022). Generative Sketch Healing. International Journal of Computer Vision, 130(8), 2006-2021.
>
>
> [F] Qi, Y., Su, G., Chowdhury, P. N., Li, M., \& Song, Y. Z. (2021). Sketchlattice: Latticed representation for sketch manipulation. In Proceedings of the IEEE/CVF International Conference on Computer Vision (pp. 953-961).
>
>
> [G] Zang, S., Tu, S., \& Xu, L. (2023). Self-Organizing a Latent Hierarchy of Sketch Patterns for Controllable Sketch Synthesis. IEEE Transactions on Neural Networks and Learning Systems.

---

> > ### Comment · Reviewer_ihan · 2023-11-22
> > **Feedback on author feedback**
> >
> > I thank the authors for their response. However, since authors themselves identify in their feedback many places where they would need to update the manuscript (no only to me, but to other reviewers as well), I feel that the manuscript is in need of major revision. Combined with the fact that authors need to come up with an "for example" example application for their proposed approach, I am not updating my rating.

---

### Official Review · Reviewer_1sUL · 2023-11-01

**Soundness:** 2 fair
**Presentation:** 3 good
**Contribution:** 2 fair
**Rating:** 3
**Confidence:** 4

**Summary:**

The paper proposes a method to perform edits on a sketch at the stroke level.

The proposed method uses denoising diffusion to convert noise into the location where each stroke is placed.
Each stroke in turn is normalized to a consistent starting location and encoded by a stroke-encoder.

After the denoising process, a decoder takes the locations and the embeddings to reconstruct the sketch.

At inference, if a chosen stroke is edited, the edit should transfer to the whole sketch as well after the deneoising process is complete.

The claimed contributions are:
1. **First** sketch synthesis method that works on the stroke level
2. A **fresh** view on the sketch synthesis problem by reformulating it as a problem of placement of strokes.
3. SOTA performance on a) sketch-reconstruction

**Strengths:**

1. The paper is easy to read.
2. The introduction is well-motivated and the related work section appropriately covers a lot of the previous literature.
3. Figure 2 gives a strong overview of the whole proposed method.

**Weaknesses:**

I find many weaknesses in the whole paper, which I will describe below:

**(MAJOR) Using the original locations for computing reconstruction metrics**

---

In Table 2. you measure the reconstruction quality of the model - this is a very unfair competition to the rest of the methods - those methods do not have access to the original locations, so it seems a bit intuitive that your method will perform really well.

What youre measuring then is then basically how well the sketch-encoder and the sequence decoder work. In practice, you have almost trained an autoencoder model!.

Without using the original location (table 1), the proposed method has a REC score on par with SP-gra2seq (Table 2). The FID is lower than other methods (because you are using original locations and/or a reconstruction loss). The LPIPS using the generated locations is comparable to Sketch-RNN which is a paper from 2017!.

Just having good reconstruction does not make a useful contribution.

**(MAJOR ) Lack of convincing qualitative comparisons**

---

The low scores would still be acceptable if the actual use-case of editing was comprehensively shown, but we are only shown a few examples in Figure 1. (where the original stroke and the edited stroke themselves aer not shown). I am not sure what the authors intend to show with the two generations - is the model unable to copy the edited stroke faithfully? or are the two generations because of two input edits? The figure and the caption do not make it clear.

The examples in Fig3 are also unconvincing - when using the locations from the diffusion model, the unedited parts change significantly!
eg the wings on the bumblebee example columns 5/6
the cat moves up and down considerably.

**(MAJOR) Because of points 1 and points 2, the novelty is unconvincing**

I would accept the paper if the results were impressive. However, they are not. This makes the paper a bit problematic to accept because the major components already exist

1. GMM modeling comes from [1] and [2] with the EM algorithm also lifted from there
2. Token based MLPs come from [3]

The major contribution would be the token mixture block but that is not presented or ablated in great detail. The mixture block is what gives the model permutation invaraince which is needed for handling sketches composed of strokes. The sequence composition is itself not detailed - do you permute the sequence of strokes or is the model itself permutation invarianct becuase of the MLP, and the encoding CNN being applied independently.

**(MINOR) Grammar not being in the continuous tense**

---

A lot of the text is titiled something like "Pre-train the stroke encoder..." and "Train the diffusion model" instead of "Pre-training the stroke encoder..." and "Training the diffusion model".


[1]: Sicong Zang, Shikui Tu, and Lei Xu. Controllable stroke-based sketch synthesis from a self- organized latent space. Neural Networks, 137:138–150, 2021.
[2]: Sicong Zang, Shikui Tu, and Lei Xu. Self-organizing a latent hierarchy of sketch patterns for con- trollable sketch synthesis. IEEE Transactions on Neural Networks and Learning Systems, 2023a.
[3]: Ilya O Tolstikhin, Neil Houlsby, Alexander Kolesnikov, Lucas Beyer, Xiaohua Zhai, Thomas Un- terthiner, Jessica Yung, Andreas Steiner, Daniel Keysers, Jakob Uszkoreit, et al. Mlp-mixer: An all-mlp architecture for vision. Advances in neural information processing systems, 34:24261– 24272, 2021.

**Questions:**

I asked all the questions in the weakness section.

---

> ### Author Response · Authors · 2023-11-17
> **Thanks for your comment!**
>
> Thank you for your feedback.
>
>
> **Q1: Using the original locations for computing reconstruction metrics. The use of original stroke locations leads to unfair comparisons.**
>
> A1:
> Our comparison is **completely fair** and the reasons are as follows. All comparison methods require a full sketch as input. The original locations of the strokes are already included in the input under two ways of representations, i.e.,
> 1) When a sketch is rasterized into an image, every stroke is drawn precisely onto the canvas at a specific location. This means that the original positional information is preserved and not discarded.
> 2) A sketch sequence consists of strokes each with a specified number of points represented as either absolute or offset coordinates, ensuring the original positional information remains intact.
>
>
> Our approach breaks down the sequence into several strokes, with shapes and positions separated to implement sketch editing. The original stroke locations are used in the decoding phase for sketch reconstruction or generation. Therefore, our method uses the same information of the original stroke locations as other methods.
>
> **Q2: Measurement of encoder and decoder performance.**
>
>
> A2: The performance of the encoder and decoder are assessed as it greatly influences the reconstruction of sketches, which in turn affects the editing of sketches. Figure 5 in the manuscript shows that previous methods for sketch reconstruction have these issues: 1) Unrecognizable reconstructed sketches. 2) Sketch category changes, and 3) Instances maintain the same category but have a significantly different shape. These issues make the editing very difficult.
>
>
> **Q3: The performance of sketch reconstruction with generated locations.**
>
>
> A3: When evaluating the reconstruction performance, our method uses the generated locations for the strokes, whereas the previous methods use the original, ground-truth stroke locations. Although the setting is in favour of the previous methods, the overall performance of our method still outperforms that of all competitors. This is due to the fact that our encoder efficiently extracts the features of the stroke shape to generate reasonable positions. When the comparison is fair, i.e., our method employs the original stroke locations the advantages of the proposed SketchEdit are significant.
>
> **Q4: Problems with Figures in the Manuscript.**
>
>
> A4: We will refine the caption of Figure 1 in our further version as follows: ``(Arrow left) Original sketches. (Arrow right) Edited sketches generated by our model. A stroke is added to the fish, the body of a cat is replaced by a sheep’s body, and the ears of a pig are replaced by the wings of an angel. The edited strokes are from the QuickDraw dataset. "
>
>
> In Figure 3, Changes in bees during interpolation and the up-and-down movement of the cat are due to the fact that the network generates editing sketches with a certain amount of randomness in the results. This is partly because the inverse process of the diffusion model requires the initialization of Gaussian noise, and partly because the output of the sequence decoder is the parameters of a Gaussian mixture model (this modeling approach is widely used [A], [B], [C]) rather than a deterministic result, which needs to be sampled in order to obtain the final result.
>
>
> **Q5: The major components already exist and the major contribution.**
>
>
> A5: The aim of our research is to present a novel framework that enables the flexible of sketch editing at the stroke-level for the first time, rather than a specific component to improve the quality of sketch generation. We also provide a new perspective on how a number of simple unlabeled strokes can be put together like building blocks to create a meaningful sketch. To illustrate why the proposed method can be used for sketch editing, the sketch reconstruction task is used to show that SketchEdit overcomes the shortcomings of the previous sketch generation methods and is able to effectively maintain the visual cues of the original sketch.
>
>
> **Q6: Grammar not being in the continuous tense.**
>
>
> A6: We thank the reviewers for pointing out grammatical errors, which will be addressed in the manuscript.
>
> [A] Qi, Y., Su, G., Wang, Q., Yang, J., Pang, K., \& Song, Y. Z. (2022). Generative Sketch Healing. International Journal of Computer Vision, 130(8), 2006-2021.
>
>
> [B] Qi, Y., Su, G., Chowdhury, P. N., Li, M., \& Song, Y. Z. (2021). Sketchlattice: Latticed representation for sketch manipulation. In Proceedings of the IEEE/CVF International Conference on Computer Vision (pp. 953-961).
>
>
> [C] Ha, D., \& Eck, D. (2018, February). A Neural Representation of Sketch Drawings. In International Conference on Learning Representations.

---

> > ### Comment · Reviewer_1sUL · 2023-11-22
> > **Received response**
> >
> > I thank the authors for the responses.
> >
> > I am however not convinced by the responses. Other reviewers also correctly point out that the reconstruction quality is not the metric by which this paper should be evaluated, and perhaps a well conducted user study is better (if not complementary).
> >
> > We would also need to look at more qualitative results than are currently presented.
> >
> > I therefore, retain my rating.
> >
> > ---
> >
> > Q5: The major components already exist and the major contribution
> >
> > The answer to this is actually fair in the sense that the overall framework is the contribution, but given the weak evaluation is still not enough to clear the bar for publication.